



# Contributions of lightning to long-term trends and inter-annual variability in global atmospheric chemistry constrained by Schumann Resonance observations

Xiaobo Wang[1,2,3], Yuzhong Zhang[2,3], Tamás Bozóki[4,5], Ruosi Liang[1,2,3], Xinchun Xie[1,2,3], Shutao Zhao[1,2,3], Rui Wang[2,3], Yujia Zhao[1,2,3], and Shuai Sun[1,2,3]

[1]College of Environmental and Resource Sciences, Zhejiang University, Hangzhou, Zhejiang, China
[2]Key Laboratory of Coastal Environment and Resources of Zhejiang Province, School of Engineering, Westlake University, Hangzhou, China
[3]Institute of Advanced Technology, Westlake Institute for Advanced Study, Hangzhou, China
[4]HUN-REN Institute of Earth Physics and Space Science, Sopron, Hungary
[5]Department of Geophysics and Space Science, Institute of Geography and Earth Sciences, ELTE Eötvös Loránd University, Budapest, Hungary

*Correspondence to*: Yuzhong Zhang (zhangyuzhong@westlake.edu.cn)

**Abstract.** Lightning is a significant source of nitrogen oxides ($NO_x \equiv NO + NO_2$) in the free troposphere. Variations in global lightning activity influence long-term trends (LTT) and inter-annual variability (IAV) in tropospheric $NO_x$, ozone ($O_3$) and hydroxyl radicals (OH). However, accurately quantifying these impacts is hindered by uncertainties in representing year-to-year fluctuations of global lightning activity in models. Here, we apply Schumann Resonance (SR) observations, which are sensitive to changes in global lightning activity, to better constrain inter-annual variations in lightning $NO_x$ ($LNO_x$) emissions. By integrating this update into an atmospheric chemical transport model, we assess the contributions of lightning to both LTT and IAV in global atmospheric chemistry from 2013 to 2021. The updated parameterization predicts an insignificant trend in global $LNO_x$ emissions, contrasting with a significant increase of 6.4% $dec^{-1}$ ($P < 0.05$) by the original parameterization, reducing lightning contributions to LTT in $NO_x$, $O_3$, and OH. The updated simulation better aligns with satellite-observed trends in global and Northern Hemispheric $NO_2$, but further underestimates tropospheric $O_3$ increases. The updated parameterization reveals twice the IAV in global $LNO_x$ emissions but 20% smaller IAVs in global $O_3$ and OH, because lightning generally counteracts other sources of natural variability. A ~10% decline in lightning in 2020 relative to 2019 led to ~2% decrease in global OH, explaining half of observed annual methane growth. These findings highlight the value of Schumann Resonance observations in constraining global lightning activity, thereby enhancing our understanding of lightning's role in atmospheric chemistry.



## 1 Introduction

Lightning generates reactive nitrogen oxides ($NO_x \equiv NO + NO_2$) by breaking down $N_2$ and $O_2$ in the intense heat of its flash channels, making it a primary natural source of $NO_x$ in the atmosphere (Schumann and Huntrieser, 2007). Although lightning accounts for only approximately 10% of the global $NO_x$ emissions, lightning $NO_x$ (referred to as $LNO_x$) emissions have a disproportionately large impact on tropospheric ozone ($O_3$) and hydroxyl radicals (OH), because lightning emits $NO_x$ directly to the free troposphere where $O_3$ and OH production is more sensitive to $NO_x$ than at the surface (Liaskos et al., 2015; Murray, 2016; Murray et al. 2021; Nussbaumer et al., 2023). For example, although $LNO_x$ emissions are of similar magnitude to those from biomass burning or soils, their contribution to the tropospheric $O_3$ is about 3 times larger (Dahlmann et al., 2011). The contribution of $LNO_x$ to global atmospheric OH abundances is more than 20% (Murray et al. 2021).

Tropospheric $O_3$ and OH are important atmospheric species both chemically and environmentally. $O_3$ is a potent greenhouse gas in the upper troposphere (IPCC, 2021), and when transported to the surface, acts as an air pollutant that adversely affects the health of ecosystems and populations (Fleming et al., 2018; Unger et al., 2020). OH is the main sink for a diverse array of reduced species in the atmosphere, thereby influencing their environmental and climate impacts. For instance, even a small fluctuation in the global mean OH concentrations can have a large effect on the budget of methane ($CH_4$), the second most important greenhouse gas after $CO_2$, at the inter-annual scale (Turner et al., 2017; Rigby et al., 2017). Therefore, through regulating atmospheric chemistry, changes in global lightning activity can carry substantial implications for ecosystems, human health, and the climate.

Most atmospheric chemistry transport models (CTMs) rely on sub-grid parameterizations of lightning flash rates to estimate $LNO_x$ emissions, as the fine-scale lightning physics cannot be fully captured by coarse-resolution CTMs (Murray et al., 2012). These parameterizations are often based on proxies of deep convection such as the cloud-top height (Price and Rind, 1992) and the upward cloud ice flux (Finney et al., 2014). The cloud-top height parameterization and its variants are commonly employed in CTMs. While the parameterized lightning flash rates align with convection patterns in simulations, they may not replicate the global spatial patterns of lightning activity (Tost et al., 2007). This limitation is improved by incorporating lightning observations from satellite-based the Lightning Imaging Sensors (LIS) and Optical Transient Detector (OTD) to constrain the distribution of $LNO_x$ emission provided by the parameterization (Murray et al., 2012).

In addition, there is no assurance that parameterized lightning flash rates can capture year-to-year variations in global lightning activity and $LNO_x$ emissions (Clark et al., 2017). Studies have reported substantial disagreement between different parameterizations on the magnitude and even the direction of LTT. For instance, parameterization based on the cloud-top height tends to predict an increase in global lightning activity in response to climate change, while estimations based on upward cloud ice fluxes predict the contrary outcome (Jacobson and Streets, 2009; Clark et al., 2017; Finney et al., 2018).



This drawback impedes the assessment of lightning's impact on the LTT and IAV of global tropospheric $O_3$ and OH budgets.

Although lightning can be observed from ground- and space-based platforms, these observations may not offer adequate global-scale constraints on changes in lightning activity. The World Wide Lightning Location Network (WWLLN), operational since 2012, is a global ground-based lightning observation network. Despite providing detailed information on
individual cloud-to-ground lightning strikes, these measurements lack sensitivity to intra-cloud lightning strikes (Abarca et al., 2010) and exhibit uneven detection efficiency globally that varies over years (Hutchins et al., 2012), thereby limiting their utility in long-term global investigations (Bozóki et al., 2023). On the other hand, satellite observations, including the OTD onboard the OrbView-1 (formerly MicroLab-1) (May 1995 to February 2000; coverage between ±70° latitude) and the LIS onboard the Tropical Rainfall Measuring Mission (TRMM) satellite (January 1998 to March 2015; coverage between
±38° latitude) and the International Space Station (ISS) (March 2017 to October 2023; coverage between ±55° latitude) provide more uniform detection. However, these satellite data records are not continuous, and their data coverage and detection efficiency vary with instrument and observation platform (Blakeslee et al., 2020; Clark et al., 2017).

Alternatively, global lightning activity can be derived from measurements of Schumann resonances (SR), which are low-frequency electromagnetic waves within the atmosphere primarily generated by lightning activity and resonating between the
surface and the ionosphere (Balser and Wagner, 1960; Schumann, 1952; Nickolaenko and Hawakawa, 2002). Despite offering limited spatial information, the SR signal is sensitive to global lightning activity, in contrast to other observations with restricted coverage and variable detection efficiencies. This global sensitivity can be attributed to the minimal attenuation of lightning-induced radio waves within this specific frequency band (Chapman et al., 1966). Long-term SR measurements are also influenced by the solar cycle, as variations in solar X-ray flux and precipitating energetic particles
lead to changes in wave propagation conditions (Bozóki et al., 2021). To effectively characterize global lightning activity using SR data, it is essential to eliminate the influence of solar cycles (Bozóki et al., 2021).

In this study, we will apply SR observations to constrain global lightning activity and $LNO_x$ emissions. Using model simulations, we will quantify the impact of this update on $LNO_x$ emissions and subsequently on global atmospheric chemistry. We focus the analysis on the global and hemispheric LTT and IAV of $NO_x$, $O_3$, and OH. We also explore the
implications of lightning variability for the global budgets of tropospheric $O_3$ and $CH_4$.

## 2 Data and Methods

### 2.1 Lightning observation data

We use SR measurements at the Eskdalemuir Geoagnetic Observatory (ESK) in the UK operated by the British Geological Survey (Beggan and Musur, 2018; Musur and Beggan, 2019) between 2013 and 2021. The measurements are performed at a
cadence of 100 Hz with two orthogonal horizontal induction coils. Following Bozóki et al. (2021), we apply the weighted





average method to the intensities of the first three SR modes, and manually exclude the disturbed days (usually due to manmade activity) from the dataset. We then derive the SR intensity, as a proxy for monthly global lightning activity, by summing the intensities of the first three resonance modes for both horizontal magnetic field components. We employ a third-order polynomial fit on the full data record to remove the influence of solar cycle variations.

To mitigate the impact of changes in the source-observer distance (Sátori et al., 2009) and day-night asymmetry (Melnikov et al., 2004), we utilize the SR intensity to parameterize the interannual variation of global lightning activity for each individual month. The premise is that in a given month the day-night condition does not change substantially and that the source-observer distance is the same year by year (Sátori et al., 2024). As a result, we use this SR dataset to estimate the shift in global lightning activity between, for instance, January 2013 to January 2014, while it is not used to discern the changes

from January 2013 to February 2013. Figure 1 presents the 2013–2021 annual variation of monthly global lightning activity derived from SR observations.

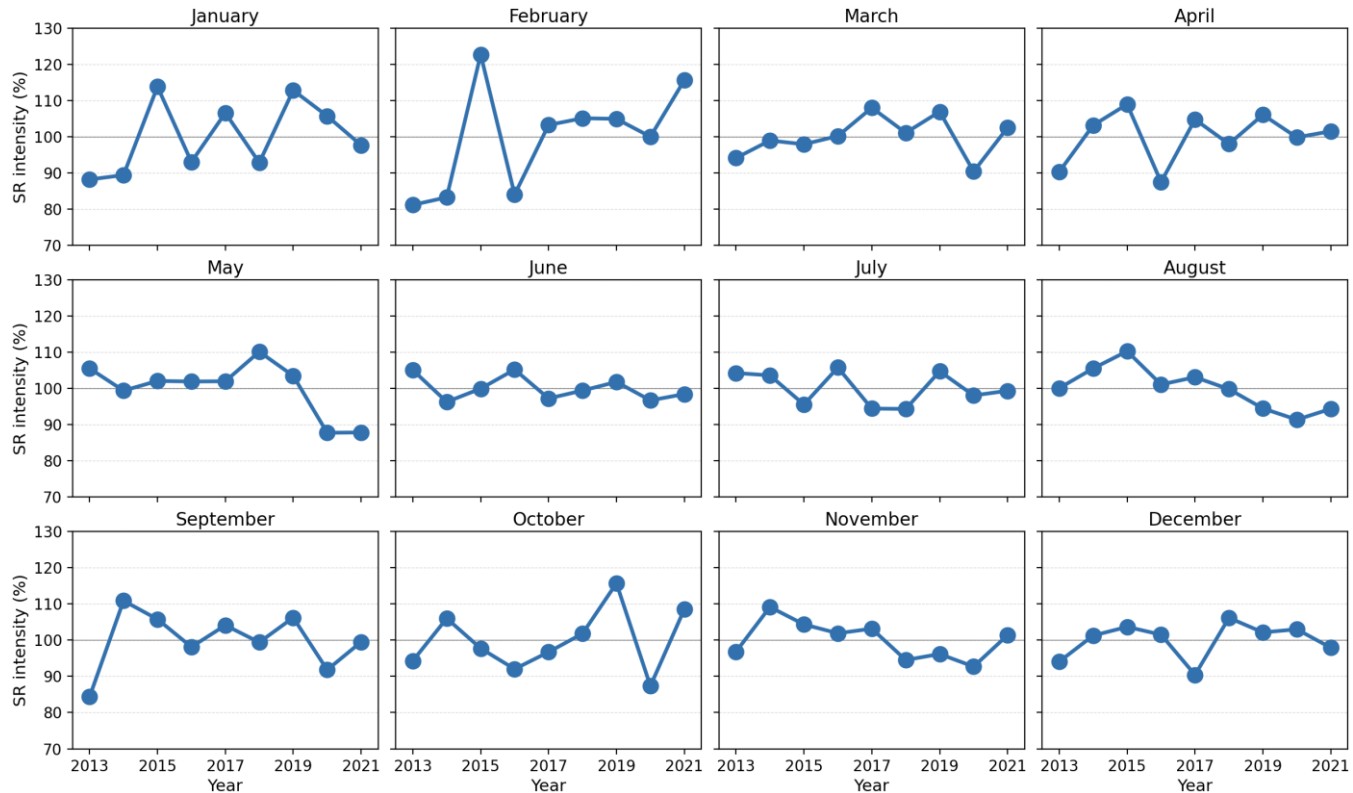

**Figure 1: The interannual variation of global lightning activity based on SR observations in different months during 2013-2021.**

We also compare SR-based global lightning activity with other global lightning observation data, including the ISS LIS data

for 2018–2021, the TRMM LIS data for 2013–2014, and the WWLLN data (v2022.0.0) for 2013–2021 (Kaplan and Lau, 2021, 2022).



## 2.2 Lightning NO$_x$ emissions

LNO$_x$ emissions are computed in GEOS-Chem as the product of the lightning flash rate ($F$) and NO production per flash ($P_{NO}$) for each grid-cell column at every model time step. The column total LNO$_x$ emissions are then vertically distributed based on Ott et al. (2010). GEOS-Chem estimates the lightning flash rate based on the cloud height parameterization (Price and Rind, 1992), with its global distribution constrained by the climatology of OTD/LIS observations (Murray et al., 2012). This method is referred to as the original parameterization (OP) hereafter.

To constrain the IAV of global lightning flash rate with SR intensity, we apply a global scaling factor ($S(m, y)$), as a function of month $m$ and year $y$, to the lightning flash rates computed from the OP method. Here we assume that for each individual month the global lightning flash rate is proportional to the SR intensity, which is consistent with Boldi et al. (2018), Dyrda et al. (2014), and Heckman et al. (1998). The scaling factor is computed as

$$S(m, y) = \frac{\overline{F_{OP}}(m)}{F_{OP}(m, y)} \frac{SR(m, y)}{\overline{SR}(m)}, \tag{1}$$

where $F_{op}(m, y)$ is the global lightning flash rate in month $m$ and year $y$ computed by the OP method, $\overline{F_{OP}}(m)$ is the multi-year (2013–2021) average of $F_{op}(m, y)$, $SR(m, y)$ is the SR intensity in month $m$ and year $y$, and $\overline{SR}(m)$ is the multi-year average of $SR(m, y)$. Hereafter, this updated parameterization is referred to as the SR method (SR).

In addition, we also update the value of $P_{NO}$ to be 330 moles NO per flash all over the world (Luhar et al., 2021), consistent with the global average derived from satellite NO$_2$ observations (Miyazaki et al., 2014; Marais et al., 2018). By comparison, the default $P_{NO}$ value in GEOS-Chem is 500 moles NO per flash north of 35° N and 260 for the rest of the world (Murray et al., 2012). Such a large enhancement of $P_{NO}$ in northern midlatitudes relative to the rest of the world is not supported by a recent analysis of observed NO$_2$ vertical profiles (Horner et al., 2024).

## 2.3 Atmospheric chemistry simulations

We use GEOS-Chem v14.0.2 in a global simulation of atmospheric chemistry between 2013 and 2021. The model includes a detailed mechanism for the O$_3$-NO$_x$-CO-VOC-aerosol chemistry with recent updates in halogen chemistry, isoprene chemistry, and aerosol and cloud NO$_x$ uptake (Wang et al., 2021; Bates and Jacob, 2019; Holmes et al., 2019). The simulation, conducted at a 2° × 2.5° horizontal resolution, is driven by reanalyzed meteorology from the Modern-Era Retrospective analysis for Research and Applications, version 2 (MERRA-2) (Gelaro et al., 2017).

The simulation uses anthropogenic emissions compiled from multiple global and regional inventories. This includes global emissions from the CEDS emission inventory (Hoesly et al., 2018), which are superseded by NEI-2011 (2011 National Emissions Inventory) in the US, APEI (Air Pollutant Emission Inventory) in Canada, MIX-Asia v1.1 in Asia (Li et al., 2017), and DICE-Africa in Africa (Marais and Wiedinmyer, 2016). Aircraft emissions for 2019, as provided by the Aircraft



Emissions Inventory Code (AEIC 2019) (Simone et al., 2013), are adjusted annually based on the Global Aviation emissions Inventory (GAIA) (Teoh et al., 2023) to account for inter-annual changes (Lee et al., 2021).

The simulation also incorporates a variety of natural emissions including fire emissions from the Global Fire Emissions Database (GFED4; van der Werf et al., 2017), biogenic emissions from the Model of Emissions of Gases and Aerosols from 140 Nature version 2.1 (MEGAN2.1; Guenther et al., 2012), soil $NO_x$ emissions from the Berkeley-Dalhousie Soil $NO_x$ Parameterization (Hudman et al., 2012), and $LNO_x$ emissions from either OP or SR methods.

We compare atmospheric chemistry simulations driven by OP- or SR-based $LNO_x$ emissions, with a focus on the LTT and IAV in $NO_x$, $O_3$, and OH. To isolate the lightning contribution to the LTT and IAV, we also perform a simulation driven by the multi-year (2013–2021) average $LNO_x$ emissions and analyze the difference from simulations with annually varying 145 $LNO_x$ emissions. The simulation with 2013–2021 average $LNO_x$ emissions also provides information on the contribution of non-lightning factors (e.g., fires, biogenic emissions, and meteorology) to the LTT and IAV in $NO_2$, $O_3$, and OH, allowing the decomposition of the LTT and IAV into the contributions of lightning and non-lightning factors.

## 3 Results and Discussion

### 3.1 LTT and IAV of global lightning activity and $LNO_x$ emissions

Figure 2 presents global lightning activity and $LNO_x$ emissions as determined by the SR method, which are compared with results from the OP method as well as lightning observations from the ground WWLLN network and satellite LIS instruments. The SR method indicates an insignificant LTT in global lightning activity and $LNO_x$ emissions over the study period, in contrast to the OP method, which predicts an increase by 6.4% dec$^{-1}$ ($P < 0.05$) from 2013 to 2021 (Fig. 2a). As a comparison, the WWLLN network also observe insignificant LTT, although the SR method and WWLLN observations do 155 not always agree on annual anomalies. Indeed, the OP method in GEOS-Chem predicts significant increases in global $LNO_x$ emissions over an even longer period (1990–2022; Fig. 3). Although our SR data are unavailable for an earlier period, this result of the OP method is again inconsistent with TRMM LIS observations, which show no trend for 1998–2014 in the tropics (Fig. 3). These results highlight the significant uncertainties in the unconstrained OP method in estimating the LTT of global lightning activity, demonstrating the value of applying the SR method to long-term atmospheric chemistry simulations.



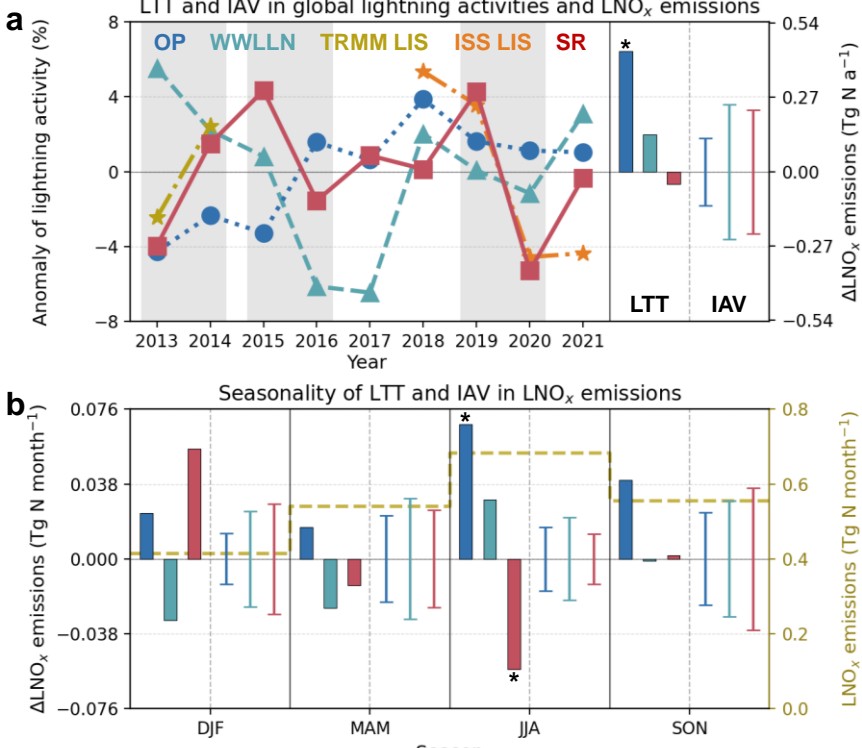

**Figure 2: (a) Annual anomalies in global lightning activities and LNOₓ emissions during 2013–2021 based on the OP method and observations from SR, WWLLN, TRMM LIS, and ISS LIS. Gray shadings indicate years with large anomalies (> 5%) in global lightning activities and LNOₓ emissions based on SR observations. LTT and IAV of the annual anomaly times series are shown to the right. (b) Seasonal LTT and IAV in global LNOₓ emissions based on the OP method and observations from SR and WWLLN during 2013–2021. Yellow dashed lines show global mean LNOₓ emissions in different seasons. LTT values represent linear changes in a decade. Asterisks indicate significant trends (*P* < 0.05). IAV values represent standard deviation of annual anomalies after LTT is removed.**





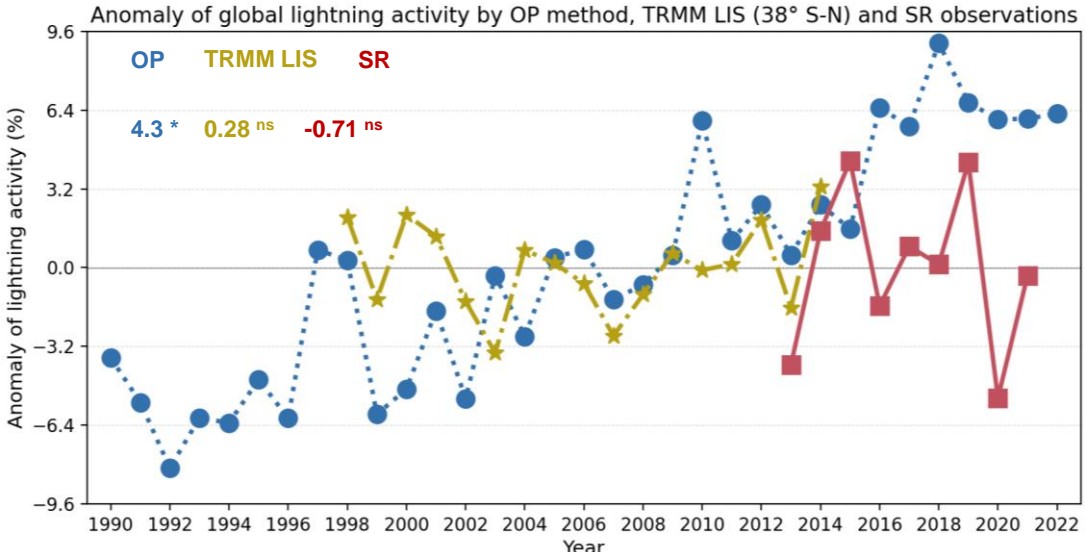

**Figure 3: Anomaly of global lightning activity based on OP method, TRMM LIS (38°S–38°N), and SR observations. The decadal**
**trends are shown in the upper left space. * denotes significant trends ($P < 0.05$) and $^{ns}$ insignificant trends.**

We quantify the magnitude of IAV in $LNO_x$ emissions by computing the standard deviation of detrended annual anomalies.
The results show that the IAV derived from the SR method (0.23 Tg N $a^{-1}$ or 3.3%) is twice that from the OP method (0.13
Tg N $a^{-1}$ or 1.8%) (Fig. 2a), indicating that the IAV in $LNO_x$ emissions is underestimated by the OP method. The magnitude
of IAV in global lightning activity observed by the WWLLN network (0.25 Tg N $a^{-1}$ or 3.6%) is close to that by the SR
observations (Fig. 2a). Consistent with our result, a previous study by Murray et al., (2013) also found that the cloud-top
height parameterization underestimates the IAV in tropical $LNO_x$ emissions by about a factor of 2 compared to TRMM LIS
satellite observations.

Seasonally, the SR method finds that the magnitudes of IAVs in $LNO_x$ emissions are 0.028 Tg N $month^{-1}$ in DJF (austral
summer) and 0.013 Tg N $month^{-1}$ in JJA (boreal summer) (Fig. 2b), which translates to relative IAVs of 6.7% and 1.8% for
the two seasons, respectively (average $LNO_x$ emissions are respectively 0.42 and 0.73 Tg N $month^{-1}$ in DJF and in JJA) (Fig.
2b). Because of the dominant occurrence of lightning in the summer hemisphere, this result based on the SR method
indicates a factor of 3.7 greater relative IAVs in $LNO_x$ emissions over the Southern Hemisphere than the Northern
Hemisphere. In contrast, the OP method predicts similar levels of relative IAVs in both hemispheres (Fig. 2b).

In terms of year-to-year anomalies, the SR method shows large increases in global lightning activity and $LNO_x$ emissions
during 2013–2015, 2018–2019 and 2020–2021 and decreases in 2015–2016 and 2019–2020 (Fig. 2a). These large year-to-
year anomalies are not captured by the OP method (Fig. 2a). Although the SR method and WWLLN observations show
consistent increases for 2020–2021 and decreases for 2015–2016 and 2019–2020, they disagree on year-to-year changes
during 2013–2015, 2017–2018, and 2018–2019, probably because of variable global detection efficiency by the ground



WWLLN network (Hutchins et al., 2012). The SR results agree well with satellite-based LIS observations for the increase
during 2013–2014 and the decrease during 2019–2020, while satellite-based observations are unavailable for 2015–2017.
Noteworthily, SR and ISS LIS observations conformably show that global lightning activity occurs in a decrease by ~10% in
2020 (Fig. 2a). Such a large inter-annual deviation is unprecedented in the history of LIS observation records (Fig. 3), which
is in accordance with previous finds by Dunn et al. (2022; see Fig. SB2.1b).

**3.2 Lightning-driven LTT of global atmospheric chemistry**

Figure 4 shows the contribution of $LNO_x$ emissions, as determined by the OP or SR methods, to the global LTT and IAV in
key atmospheric chemicals ($NO_x$, $O_3$, and OH), based on GEOS-Chem simulations. The contribution of lightning is further
compared to those of other influencing factors. The stable $LNO_x$ emissions predicted by the SR method result in insignificant
lightning-induced LTT in $NO_x$, $O_3$ and OH during 2013–2021, both globally and in two hemispheres (Fig. 4a-c, Fig. S1a-c
and Fig. S2a-c). In contrast, the OP method, with increases in $LNO_x$ emissions, leads to a substantial lightning-induced LTT
in these species. These trends are statistically significant for $NO_x$ (2.1% dec$^{-1}$), $O_3$ (1.0% dec$^{-1}$), and OH (1.4% dec$^{-1}$) in the
Northern Hemisphere and for $NO_x$ (1.9% dec$^{-1}$) globally (Fig. S1a-c and Fig. 4a).




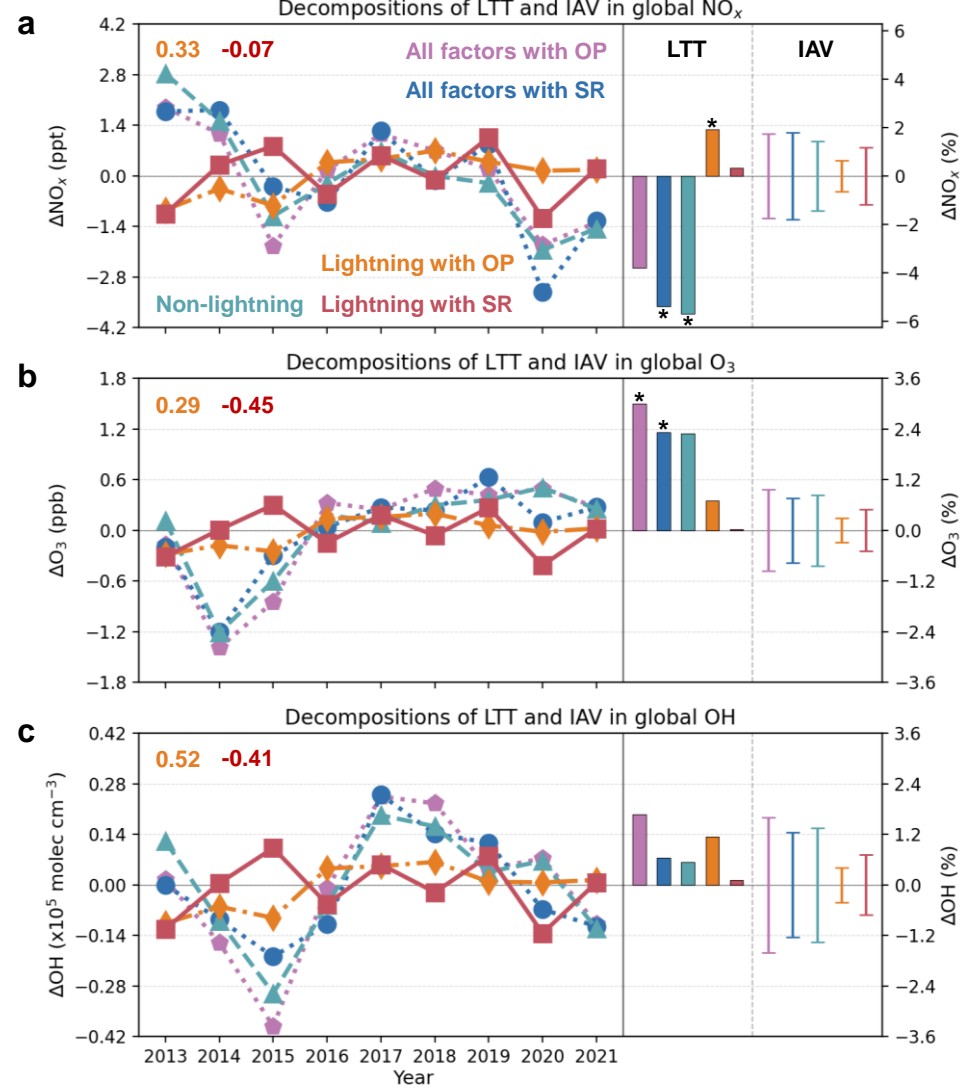

**Figure 4: Annual anomalies, LTT, and IAV of global $NO_x$ (a), $O_3$ (b) and OH (c), and their decompositions into lightning and non-lightning factors during 2013–2021. Values for LTT represent linear changes in a decade. Asterisks indicate the trend is significant ($P < 0.05$). IAV values represent standard deviation of annual anomalies after LTT is removed. Correlation coefficients between the lightning (OP: orange; SR: red) and non-lightning contribution are denoted in the upper left of each panel.**

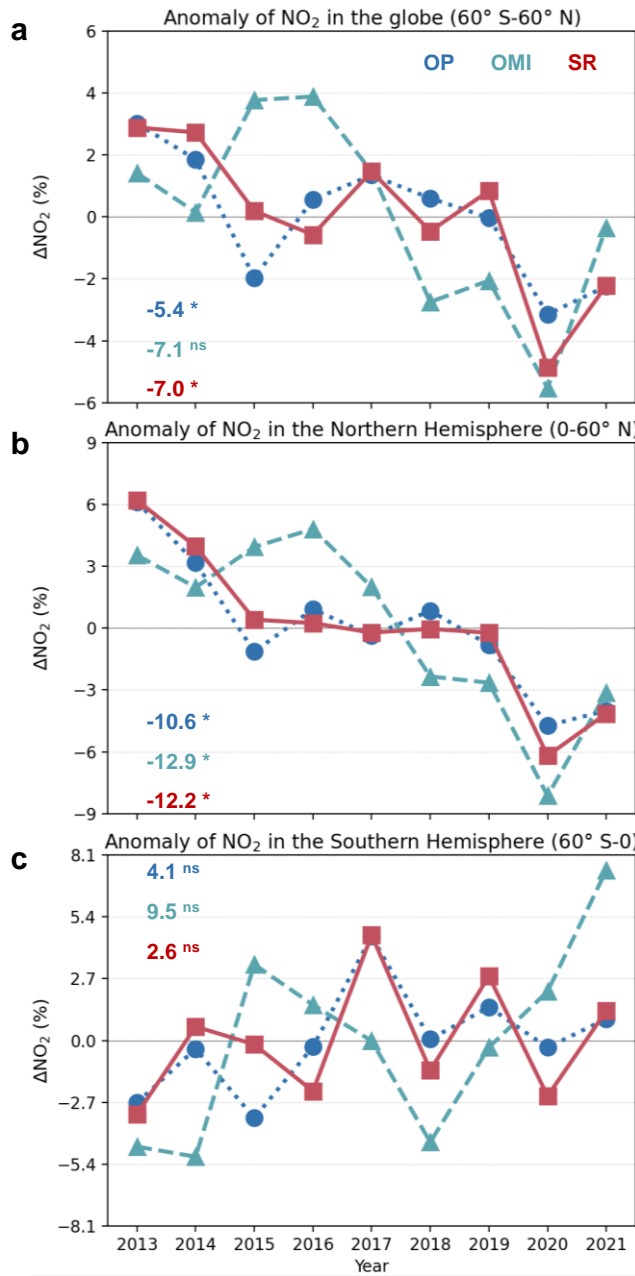

**Figure 5: The anomaly of tropospheric NO$_2$ vertical column densities averaged over the globe (a), Northern Hemisphere (b) and Southern Hemisphere (c) during 2013-2021 for OP and SR simulations and OMI satellite observations. The decadal trends are inset. * represents significant trends ($P < 0.05$) and $^{ns}$ insignificant trends. Following Qu et al. (2021), we filter the data using quality flags reported in OMI data product. We exclude data affected by the row anomalies and retain those with cloud fraction < 0.2, surface albedo < 0.3, solar zenith angle < 75°, and view zenith angle < 65°.**

When all factors are accounted for, the GEOS-Chem model based on the SR method simulates a strong decrease in global (-5.4% dec$^{-1}$) and Northern Hemispheric (-10% dec$^{-1}$) NO$_x$ concentrations during 2013–2021 (Fig. 4a and Fig. S1a), driven



primarily by reduction in global anthropogenic emissions especially from China (Wang et al., 2022). However, if the simulation is based on the OP method, these negative trends are reduced to -3.8% dec$^{-1}$ globally and -8.3% dec$^{-1}$ in the Northern Hemisphere (Fig. 4a and Fig. S1a). The sharper decrease predicted by the SR simulation is in better agreement with the trends of tropospheric $NO_2$ columns from satellite observations during the period (Fig. 5a-b). This difference between the OP- and SR-based simulations highlights the critical role of accurately accounting for lightning in better interpreting the long-term trend of global background $NO_x$ levels and estimating the expected effects of emission reduction policies for improving air quality.

The GEOS-Chem simulation using SR $LNO_x$ emissions estimates ~2.5% dec$^{-1}$ increases for global and hemispheric tropospheric $O_3$ (Fig. 4b, Fig. S1b, and Fig. S2b). These trends are ~30% greater when the simulation is based on the OP method. This result suggests that non-lightning factors are major drivers for the simulated increases in tropospheric $O_3$, while lightning is a non-negligible contributor. Previous studies have reported that CTMs tend to underestimate the observed long-term increase in tropospheric $O_3$ (Christiansen et al., 2022; Wang et al., 2022), for a longer timeframe (since the 1990s) than our current study. Their results are based on $LNO_x$ emissions by the OP method, which predicts accelerated increases in global lightning activity in last three decades (4.3% dec$^{-1}$ increase during 1991–2000, 6.9% dec$^{-1}$ during 2001–2010, and 7.9% dec$^{-1}$ during 2011–2020), inconsistent with the SR intensity in our study and satellite observations for earlier period (Fig. 3). This result implies that after adjusting the LTT of $LNO_x$ emissions with SR observations, the model underestimation of $O_3$ trends should be even more pronounced, underscoring the gap in our understanding of tropospheric $O_3$ trends.

### 3.3 Lightning-driven IAV in atmospheric oxidants and implications for the global methane budget

Figures 4b and 4c show that the simulation using SR $LNO_x$ emissions exhibits greater magnitudes of lightning-induced IAVs in global average $O_3$ and OH concentrations compared to the simulation using OP $LNO_x$ emissions. This is due to the larger IAV in $LNO_x$ emissions predicted by the SR method (Fig. 2a). However, when all factors are accounted for, the IAVs in global $O_3$ and OH is more than 20% smaller in the SR simulation compared to the OP simulation. The smaller all-factor IAVs in the SR simulation arises from a negative correlation between the lightning and non-lightning contributions (r ≈ -0.4), indicating that lightning serves as a balancing factor that reduces the IAVs of $O_3$ and OH globally. Conversely, in the OP simulation, the lightning contributions positively correlate with non-lightning contributions (r = 0.3 for $O_3$ and 0.5 for OH), thereby amplifying the IAVs of $O_3$ and OH (Fig. 4b and 4c).

The year-to-year variations in global OH concentrations can impact the annual growth rates of atmospheric methane, as the OH radical is the main sink of methane in the atmosphere. NOAA observations indicate high annual methane growth rates in 2014, 2015, 2020, and 2021 (Fig. 6), although the underlying causes remain not fully understood (Feng et al., 2023; Stevenson et al., 2022; Yin et al., 2021; Zhang et al., 2021; Zhang et al., 2023). Figure 6 quantifies the contribution of lightning-induced OH variations to annual methane growth rates from 2013 to 2021, based on the simulation using SR $LNO_x$





emissions. The influence of other factors, including direct methane emissions and OH variations due to non-lightning drivers, is deduced from the difference between observed growth rates and lightning contributions.

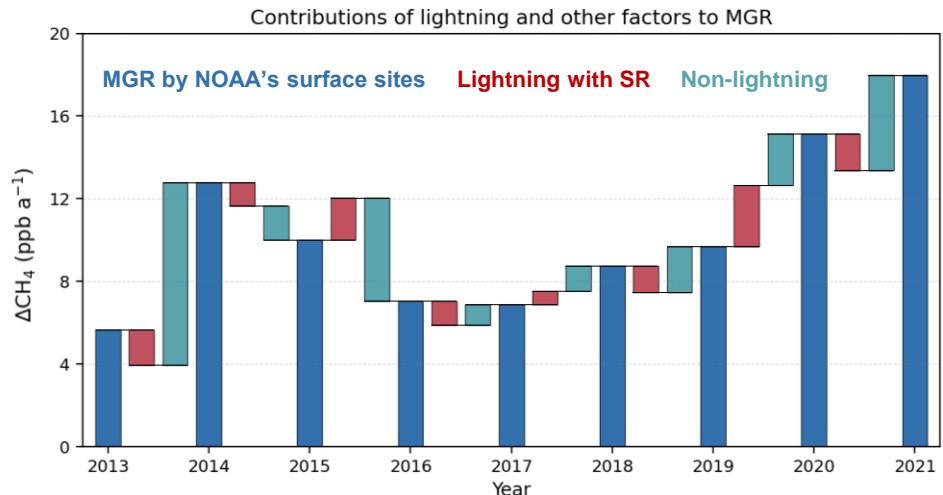

**Figure 6: Contributions of lightning-driven variations in the methane sink by OH oxidation to annual atmospheric methane growth rate (MGR) from 2013 to 2021 based on SR observations. The gap between observed MGR and simulated lightning contribution is attributed to non-lightning factors.**

Based on the simulation using SR emissions, lightning-driven OH variations account for 0.7–3.0 ppb a$^{-1}$ year-to-year anomalies in methane growth rates, with positive contributions in 2015–2016, 2017–2018, and 2019–2020 and negative contributions in other years (Fig. 6). In most cases, the influence of lightning counteracts that of other factors, suppressing changes in methane growth rates, as seen in 2013–2014, 2015–2016 and 2020–2021. However, there are also cases where lightning works in conjunction with other factors. Notably, in 2019–2020, both lightning and other factors contribute to an accelerated methane growth. As mentioned in Section 3.1, global lightning occurs in a huge decline by ~10% in 2020. Our analysis indicates that the reduction in global lightning activity can generate ~2% decrease in OH sink and explain ~54% the 2020 surge of atmospheric methane. Current theories have not accounted for the impact of abnormal lightning activity on the 2020 methane spike (Feng et al., 2023; Peng et al., 2022; Stevenson et al., 2022).

### 3.4 Comparison of year-to-year anomalies to regional observations

The integration of SR observations into the OP parameterization results in differences of several percent in the annual average global background concentrations of $NO_x$, $O_3$, and OH. While these differences between the SR and OP methods can have substantial implications for the global budgets of these species, systematic assessment using atmospheric chemical observations is challenging due to the impact of other factors across multiple scales. Here, we present two cases characterized by strong lightning-driven year-to-year anomalies, allowing us to compare simulations driven by OP and SR parameterizations to regional atmospheric observations.



In the first case, we analyze the change of high-altitude $NO_2$ concentrations over South America between September-November of 2019 and 2020, comparing the changes simulated with OP and SR $LNO_x$ emissions with those observed by the
270 tropospheric monitoring instrument (TROPOMI). The time frame is chosen due to the substantial decrease in global lightning activity detected by SR observations (Fig. S3a). South America is chosen because the continent is a lightning hotspot (Fig. S3b) and, unlike other continental hotspots, experiences lower interference of confounding factors such as fire (e.g., Africa and Indonesia) and anthropogenic (e.g., North America) emissions (see Supplementary Information for detailed discussion).

Using the cloud-slicing technique (Marais et al., 2021; Marais and Roberts, 2020) on TROPOMI observations (version 01-03-02 accessible through the Sentinel-5P Pre-Operations Data Hub), we retrieved mean $NO_2$ concentrations in the upper troposphere (300–100 hPa), where the impact of lightning is greatest (Fig. S3c), at a $2° \times 2.5°$ resolution for September-November of 2019 and 2020 (Fig. S4). 3-month cloud-sliced satellite observations over South America are averaged and compared to GEOS-Chem simulations. The analysis reveals a change in $NO_2$ concentrations of -3 [-8–3] ppb, as observed by
TROPOMI, in the upper troposphere over South America, which aligns closely with the simulation driven by SR $LNO_x$ emissions (-5.5 [-11–0] ppb) (Fig. 7a). In contrast, the simulation driven by OP $LNO_x$ emissions results in an increase in $NO_2$ concentrations (9 [4–14] ppb) (Uncertainties shown in the brackets are estimated with bootstrapping; Same hereafter).

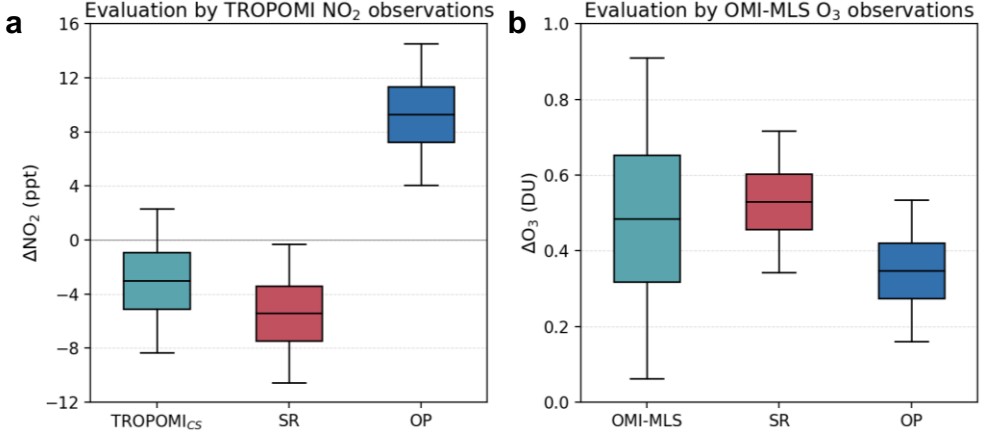

**Figure 7: (a) Anomalies of $NO_2$ concentrations in the upper troposphere (300–100 hPa) of Northern South America (20° S–10° N**
**and 80° W–40° W) in September-November 2020 relative to 2019 from the TROPOMI $NO_2$ observations by cloud-slicing technique (green), GEOS-Chem simulation with SR method (red), and GEOS-Chem simulation with OP method (blue). (b) Anomalies of tropospheric $O_3$ in the North Atlantic (0–60° N and 100° W–30° E) in 2015 relative to 2014 from the OMI-MLS $O_3$ observations (green), GEOS-Chem simulation with SR method (red), and GEOS-Chem simulation with OP method (blue). Uncertainties are calculated by 1,000,000 Monte Carlo simulations.**

In the second case, we examine changes in tropospheric $O_3$ concentrations over the North Atlantic Ocean in 2014–2015. Russo et al. (2023) highlighted that annual anomalies in tropospheric $O_3$ over this region are closely linked to lightning activity, but this variability is not well captured by atmospheric chemical models. One such instance is the substantial





increase in tropospheric ozone by 0.49 [0.1–0.9] DU measured by the OMI-MLS satellite product between 2014 and 2015 shown in Fig. S5 (Russo et al., 2023). In comparison, our GEOS-Chem simulation with OP $LNO_x$ emissions underestimates this increase (0.35 [0.2–0.5] DU), while the simulation with SR $LNO_x$ emissions better agrees with observed ozone changes (0.53 [0.3–0.7] DU) (Fig. 7b). However, we note that the difference between the OP and SR results is not significant if the uncertainties are accounted for (Fig. 7b), indicating that this case represents weak evidence supporting the SR method.

## 4 Conclusions

In this study, we updated the parameterization of $LNO_x$ emissions in GEOS-Chem model by incorporating SR observations, which offer strong global sensitivity to lightning. This update aimed to better constrain the LTT and IAV in global lightning activity and hence improve the simulation of global atmospheric chemistry.

The updated global $LNO_x$ emissions showed insignificant LTT from 2013 to 2021, while the original results predicted a significant increase by 6.4% $dec^{-1}$ ($P < 0.05$). As a comparison, the WWLLN network also observed no significant trend. Indeed, the original method in GEOS-Chem indicated significant increases in global $LNO_x$ emissions over an even longer period (1990–2022). Although our SR data were unavailable for an earlier period, this result of the original method was again inconsistent with TRMM LIS observations that shown no trend for 1998–2014 in the tropics ($\pm38°$ latitude). Meanwhile, the updated $LNO_x$ emissions had an IAV twice as high as that the original result, which was close to that by WWLLN observations. These results highlighted the significant uncertainties in the unconstrained method in estimating the LTT and IAV of global lightning, demonstrating the value of applying the SR method to long-term atmospheric chemistry simulations.

Compared to the original simulation, the simulation with updated $LNO_x$ emissions found smaller contributions of lightning to the LTT in $NO_x$, $O_3$, and OH. This led to sharper decreases in simulated global and Northern Hemispheric $NO_x$ concentrations during 2013–2021, in better agreement with the trends of tropospheric $NO_2$ columns from satellite observations. Previous researches had reported that CTMs tended to underestimate the observed long-term increase in tropospheric $O_3$ (Christiansen et al., 2022; Wang et al., 2022). Our study represented that the increase in tropospheric $O_3$ from the original simulation was ~30% greater than the updated results, implying the updated simulation further underestimated tropospheric $O_3$ trends, highlighting the gap in our understanding of tropospheric $O_3$.

Moreover, lightning-induced anomalies in $O_3$ and OH were negatively correlated with those caused by non-lightning factors (e.g., wildfires, biogenic emissions and meteorology). This suggested that lightning served as a counter-balance factor, reducing the IAV of $O_3$ and OH, a opposite feature shown in the original simulation. Our analysis showed that the change in global OH concentration due to IAV in $LNO_x$ emissions was a substantial contributor to the inter-annual variations in methane growth rates. Notably, SR and ISS LIS observations consistently found that 2020 global lightning fell by ~10%





relative to 2019 in line with Dunn et al. (2022), which might be the biggest year-to-year change in nearly three decades. It resulted in ~2% drop of OH levels, representing ~54% of the growth of atmospheric methane in this year. So, our study also

suggested that the variability in lightning activity and $LNO_x$ emissions need to be taken into account when making global atmospheric methane budget.



*Data Availability*. We used lightning observation data, including the ISS LIS data (https://ghrc.nsstc.nasa.gov/pub/lis/iss/data/science/final/nc/, last accessed 27 June 2024), the TRMM LIS data (https://ghrc.nsstc.nasa.gov/pub/lis/trmm/data/science/final/nc/, last accessed 27 June 2024), the WWLLN data (v2022.0.0)
(https://zenodo.org/records/6007052, last accessed 27 June 2024) (Kaplan and Lau, 2021, 2022) , and Schumann Resonance derived monthly lightning variation data (shown in Fig. 1). The source code of GEOS-Chem (v14.0.2) is freely available through     https://doi.org/10.5281/zenodo.7383492.     The     tropospheric     $NO_2$     observation     data     are     available     at https://disc.gsfc.nasa.gov/datasets/OMNO2_003/summary. The MGR data from NOAA's surface sites are download from https://gml.noaa.gov/ccgg/trends_ch4/. We used the cloud-slicing algorithm introduced by Marais et al. (2021) and
corresponding Python code (https://zenodo.org/records/3979211, last accessed 23 June 2024) and the improvements in the code afterwards (https://zenodo.org/records/4058442, last accessed 23 June 2024) released by Marais and Roberts (2020). The OMI-MLS observation data are download from https://acd-ext.gsfc.nasa.gov/Data_services/cloud_slice/.

*Author contributions*. Study concept by YZZ and XBW. XBW led the formal analysis and simulated GEOS-Chem with supervision from YZZ. Methodology by XBW, YZZ, TB, RSL, XCX, STZ, RW, YJZ, and SS. Resources from XBW, YZZ,
TB. Visualization by XBW. Writing – original draft by XBW; Writing – review & editing by YZZ and TB.

*Competing interests*. The authors declare that they have no conflict of interest.

*Acknowledgements*. The study is supported by the National Key Research and Development Program of China (2022YFE0209100), the National Natural Science Foundation of China (42275112), and the foundation of Westlake University. T.B.'s contribution was supported by the National Research, Development, and Innovation Office, Hungary-
NKFIH (PD146019). SR global lightning activity is based on the high frequency magnetic field induction coil data from Eskdalemuir Observatory, UK, which are supplied by Natural Environment Research Council (UK) and are available from the National Geoscience Data Centre (NGDC). We thank Ciarán Beggan (British Geological Survey) for initial processing of the induction coil dataset and valuable suggestions for enhancing the clarity of the text.



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
