# Peer review of "Atmospheric Chemistry and Physics"

_EGUsphere, 2025_

## Author Comment (AC1)

**REPLY TO REVIEWERS**

Dear editor and reviewers:

Thanks for your time and comments for our manuscript entitled "**Contributions of lightning to long-term trends and inter-annual variability in global atmospheric chemistry constrained by Schumann Resonance observations**" (Manuscript Number: EGUSPHERE-2025-370). The manuscript has certainly benefited from these insightful revision suggestions. Below we provide point-wise response to reviewers' comments. In the manuscript, revised or newly added sentences are highlighted in yellow color which relates to the reviewers' comments.

**[General comment]**

**This is an excellent paper scientifically and is also very well written. The authors use observations of the Schumann Resonance to modify the existing lightning parameterization in the GEOS-Chem model. The existing Price and Rind cloud-top height prediction scheme for flash rate, as modified by Murray et al. based on satellite lightning observations, produced an increase in flash rates over the 2013 to 2021 period. Correcting the parameterization using the Schumann Resonance yielded no significant trend, which is in line with observations. The authors used the revised scheme in GEOS-Chem simulations for the period, examining the effects of the change in lightning scheme on $NO_x$, $O_3$, and OH. The updated scheme does better at producing interannual variability in these species. The authors also examine the effects of a 10% decrease in lightning in 2020 on methane growth as a result of the decreased OH. I have some suggestions for minor changes. Once they are attended to, the paper should be ready for acceptance.**

Reply:

We thank the reviewer for constructive suggestions!

**[Main comment 1]**

**Lines 121-125: It would be worth noting that also the results of Allen et al. (2019,**

**JGR) and Bucsela et al. (2019, JGR) from use of OMI NO₂, also found little difference in LNOₓ production per flash between midlatitudes and tropics.**

Reply:

Thank you for the suggestion. We have cited these two studies to support our approach in Lines 129–130, 362–365 and 383–385, as follows:

Lines 129–130 (our setting of the $P_{NO}$ value):

In addition, we also update the value of $P_{NO}$ to be 330 moles NO per flash all over the world (Luhar et al., 2021), consistent with the global average derived from satellite $NO_2$ observations (Miyazaki et al., 2014; Marais et al., 2018). By comparison, the default $P_{NO}$ value in GEOS-Chem is 500 moles NO per flash north of 35° N and 260 for the rest of the world (Murray et al., 2012). Such a large enhancement of $P_{NO}$ in northern midlatitudes relative to the rest of the world is not supported by a recent analysis of observed $NO_2$ vertical profiles (Horner et al., 2024). NO₂ satellite observations also indicate little difference in $P_{NO}$ between midlatitudes and tropics (Allen et al., 2019; Bucsela et al., 2019).

Lines 362–365 and 383–385 (adding new references):

Allen, D. J., K. E. Pickering, E. Bucsela, N. Krotkov, and R. Holzworth (2019). Lightning NOₓ Production in the Tropics as Determined Using OMI NO₂ Retrievals and WWLLN Stroke Data, *Journal of Geophysical Research: Atmospheres*, *124*(23), 13498-13518, doi:10.1029/2018jd029824.

Bucsela, E. J., K. E. Pickering, D. J. Allen, R. H. Holzworth, and N. A. Krotkov (2019). Midlatitude Lightning NOₓ Production Efficiency Inferred From OMI and WWLLN Data, *Journal of Geophysical Research: Atmospheres*, *124*(23), 13475-13497, doi:10.1029/2019jd030561.

**[Main comment 2]**

**Figures 2, 4, 5, 6, and 7: It is difficult to tell the gray and blue bars and lines apart. A different color is needed for one of them.**

Reply:

Thanks for your suggestion. We have modified these figures following your recommendation. Below shows an example of updated Fig. 2.

[Figure]

**Updated Figure 2**

**Figure 6: This figure needs better explanation. From this figure I don't see how the non-lightning contribution is a difference between the observation and the model lightning contribution. For 2020, how do these bars imply a 54% contribution of lightning to the methane growth? Why is there a methane growth rate bar for 2021, but not the lighting and non-lightning bars?**

Reply:

Thanks for your suggestion and sorry for this confusion. We now clarify in the caption. Briefly, blue bars in Fig. 6 present annual atmospheric methane growth rates. The difference in annual growth rates between adjacent years are attributed to year-to-year changes in lightning (red) and non-lightning factors (green). For the 2019-to-2020 case, the global atmospheric methane growth rate base on observations increased by ~5.5 ppb a$^{-1}$ (from 9.7 ppb a$^{-1}$ to 15.2 ppb a$^{-1}$). Our analysis suggests that 3.0 ppb a$^{-1}$ can be

attributed to changes in global lightning activity, and thus explains 54% of the 2019-to-2020 methane surge.

[Figure]

**Figure 6: Contributions of lightning-driven variations in the methane sink by OH oxidation to annual atmospheric methane growth rate (MGR) from 2013 to 2021 based on SR observations. The differences in MGR between adjacent years are attributed to year-to-year changes in either lightning or non-lightning factors.**

**Lines 271-273: There is a significant amount of fire in South America during September to November. I think you should choose a different set of months (maybe December to February).**

Reply:

Thanks for the suggestion. Our choice was based on the relative importance of fire and lighting in these months. Although fire influence is non-negligible in September to November, the changes in lightning activity and $LNO_x$ emissions between 2019 and 2020 are most significant in these months (Figure S3a). Our simulation indicates that the effects of fire emissions on the anomaly of $NO_2$ concentrations in the upper troposphere (300–100 hPa) of Northern South America are smaller (4.5±0.1 ppt) than the effects of lightning based on SR observations (-20.1±0.5 ppt). In contrast, changes in lightning are small during December to February (Figure S3a), making it difficult for evaluating the OP and SR methods. We now add more clarification in Text S1 to explain the selection of September–November.

**[Main comment 5]**

Figure 7 - cation: "green" should be "gray".

Reply:

Thanks for your valuable suggestion. The gray color in Fig. 7 has been revised to green, as shown below.

[Figure]

**Figure 7: (a) Anomalies of NO₂ concentrations in the upper troposphere (300–100 hPa) of Northern South America (20° S–10° N and 80° W–40° W) in September-November 2020 relative to 2019 from the TROPOMI NO₂ observations by cloud-slicing technique (green), GEOS-Chem simulation with SR method (red), and GEOS-Chem simulation with OP method (blue). (b) Anomalies of tropospheric O₃ in the North Atlantic (0–60° N and 100° W–30° E) in 2015 relative to 2014 from the OMI-MLS O₃ observations (green), GEOS-Chem simulation with SR method (red), and GEOS-Chem simulation with OP method (blue). Uncertainties are calculated by 1,000,000 Monte Carlo simulations.**

---

## Author Comment (AC2)

**REPLY TO REVIEWERS**

Dear editor and reviewers:

Thanks for your time and comments for our manuscript entitled "**Contributions of lightning to long-term trends and inter-annual variability in global atmospheric chemistry constrained by Schumann Resonance observations**" (Manuscript Number: EGUSPHERE-2025-370). The manuscript has certainly benefited from these insightful revision suggestions. Below we provide point-wise response to reviewers′ comments. In the manuscript, revised or newly added sentences are highlighted in yellow color which relates to the reviewers′ comments.

**[General comment]**

**In this manuscript, Schumann resonance observations are used to create a scaling factor that further constrains the typical GEOS-Chem lightning parameterization based on cloud-top heights and the climatology of satellite observations of lightning. The trends and variability in global lightning activity, $LNO_x$ emissions, $O_3$, and OH are then compared between GEOS-chem model runs with the Schumann resonance constraint, GEOS-chem model runs without the Schumann resonance constraint, satellite lightning observations, and ground-based lightning observations. Overall, the paper is well organized, and the incorporation of Schumann resonance data is a novel approach to modeling lightning and lightning chemistry. However, there are a few questions I would like to see addressed before I recommend publication.**

Reply:

We thank the reviewer for constructive suggestions!

**[Main comment 1]**

**Line 91 states: "…and manually exclude the disturbed days (usually manmade activity) from the dataset." Could you provide more information about the disturbed days, e.g. How do you know when a day is disturbed? What are the**

**causes of disturbed days (i.e., what specific manmade activities lead to disturbances)?**

Reply:

Thanks for your question. We have modified the sentence as follows:

Following Bozóki et al. (2021), we apply the weighted average method to obtain the intensities of the first three SR modes, and manually exclude the disturbed days from the dataset. Such disturbed days are either of natural (e.g., local lightning activity, see Tatsis et al., 2021) or manmade (e.g., nearby human activity, see Tritakis et al., 2021) origin. The removal of these data is based on the evidence that the diurnal variation of SR intensities is usually very similar within a given month (Bozóki et al., 2021; Sátori 1996). Therefore, days with unusual diurnal variation are considered to be disturbed and are removed from the dataset during the manual sanitation process.

New references:

Sátori, G. (1996). Monitoring schumann resonances-11. Daily and seasonal frequency variations, *J Atmos Terr Phys*, *58*(13), 1483-1488, doi:10.1016/0021-9169(95)00146-8.

Tatsis, G., et al. (2021). Correlation of local lightning activity with extra low frequency detector for Schumann Resonance measurements, *Science of The Total Environment*, 787, doi:10.1016/j.scitotenv.2021.147671.

Tritakis, V., I. Contopoulos, C. Florios, G. Tatsis, V. Christofilakis, G. Baldoumas, and C. Repapis (2021). Anthropogenic Noise and Its Footprint on ELF Schumann Resonance Recordings, *Frontiers in Earth Science*, 9, doi:10.3389/feart.2021.646277.

**[Main comment 2]**

**Line 94: How does applying a third-order polynomial fit on the data remove the influence of solar cycle variations?**

Reply:

Thanks for your question. The following figure demonstrates how this process works:

[Figure]

The solar cycle appears as a slowly varying trend in the measured SR intensities, which is not related to lightning but to the changing propagation conditions of ELF waves (see Bozóki et al., 2021). This trend is successfully removed by the applied third-order polynomial fit.

**[Main comment 3]**

**Figures 4, S1, S2: Could you explain a little more how the correlation coefficients in these figures are determined? For example, in Fig. 4a, how is the OP lightning contribution positively correlated with the non-lightning contribution if the OP lightning LTT is positive and the non-lightning LTT is negative?**

Reply:

Thanks for your question. We now clarify in the captions that the correlation coefficients are computed between annual anomalies of lightning and non-lightning contributions, after their respective LTTs are removed.

**[Main comment 4]**

**Line 217: "The sharper decrease predicted by the SR simulation is in better agreement with the trends of tropospheric NO₂ columns from satellite observations during the period (Fig. 5a-b)." This interpretation does not seem consistent with the Figure 5 for the following reasons:**

- **Looking at Figure 5, the NO₂ anomalies determined from the SR method**

appears much more similar to those from the OP method than to the OMI observations. For example, both the OP and SR results underestimate the 2015-2016 $NO_2$ anomalies and overestimate the anomalies in 2018-2019 by about the same amounts relative to the OMI data. Thus the SR simulation does not substantially improved the agreement with the OMI data compared to the OP simulation. Are the differences between the OP and SR trends (-5.4% [SR] vs. -3.8% [OP] for global and -10% [SR] vs. -8.3% [OP] for the Northern Hemisphere) significant?

■ It is true the magnitude of the SR trend (-7.0) is closer to the OMI trend (-7.1) than to the OP trend (-5.4) in Figure 5a, but the OMI trend is not significant, while the OP and SR trends are both significant. Again, this result suggests that the SR results are not really capturing the OMI trend any better than the OP results.

Perhaps the conclusion is that some other factor besides lightning is the reason for the differences between the model results and satellite observations for the $NO_2$ anomalies?

Reply:

Thanks for your valuable comment. We agree with the reviewer that the annual anomalies of $NO_2$ are heavily affected by non-lightning factors. The mismatch between simulated and observed annual anomalies are not corrected by our update of lightning parameterization. We now clarify in the text "The sharper decrease predicted by the SR simulation is in better agreement with the trends of tropospheric $NO_2$ columns from satellite observations during the period (Fig. 5a-b), though the simulation still does not fully capture the inter-annual anomalies possibly due to errors in non-lightning factors."

In this figure, however, we intend to focus on the comparison of long-term trends, which are improved by the SR simulation. To guide the readers, we now add trend lines of the annual data in Fig. 5. We also denote significant ($P < 0.05$) and moderately significant ($0.05 \leq P < 0.1$). In the case of global OMI $NO_2$ trend, it falls into the category of "moderately significant". The observed trend is less significant compared to

simulations because of its larger inter-annual variability, which, as discussed above, is likely due to non-lightning factors that are not captured by the model simulation.

[Figure]

**Figure 5: The anomaly and linear trend of tropospheric NO$_2$ vertical column densities averaged over the globe (a), Northern Hemisphere (b) and Southern Hemisphere (c) during 2013-2021 for OP and SR simulations and OMI satellite observations. The trends are shown in the unit of % dec$^{-1}$. ** denotes significant trends ($P < 0.05$), * moderately significant trends ($0.05 \leq P < 0.1$) and $^{ns}$ insignificant trends. Following Qu et al. (2021), we filter the data using quality flags reported in OMI data product. We exclude data affected by the row anomalies and retain those with cloud fraction < 0.2, surface albedo < 0.3, solar zenith angle < 75°, and view zenith angle < 65°.**

**[Main comment 5]**

**Fig. 7a: What is causing the positive change in NO₂ anomaly for the OP method? According to Fig. S3a, LNOₓ determined from the OP method also decreases from 2019 to 2020, just not as much as the SR method LNOₓ. So it is a little surprising to see a positive NO₂ anomaly from the OP method in Figure 7a.**

Reply:

Thanks for your question. Wildfire emissions contributed ~50% of the positive change in NO₂ anomaly for the OP method, with meteorological factors explaining an additional ~30% of the variation.

**[Main comment 6]**

**Throughout the paper the trends are often described as decadal or the changes put in terms of % per decade, but the study range only covers 8 years of data, so describing the trends as decadal (occurring over 10 years) seems not quite right. Perhaps "overall" trend or similar wording would be more appropriate?**

Reply:

We now use LTT or trends throughout the text. We clarify that we report the trends in % dec$^{-1}$.

**[Technical Comment 1]**

**Line 54: "…lightning observations from satellite-based the Lightning Imaging Sensors (LIS) and Optical Transient Detector (OTD)…". The words "the" and "satellite-based" should be switched: "…lightning observations from the satellite-based Lightning Imaging Sensors (LIS) and Optical Transient Detector (OTD)…".**

Reply:

Thanks for your kind suggestion. The corresponding revision has been provided as follows:

This limitation is improved by incorporating lightning observations from ==the satellite-based== Lightning Imaging Sensors (LIS) and Optical Transient Detector (OTD) to constrain the distribution of LNOₓ emission provided by the parameterization (Murray et al., 2012).

**[Technical Comment 2]**

**Line 87: "Geoagnetic" should be "Geomagnetic" ("m" is missing).**

Reply:

Thanks for your valuable suggestion. This mistake has been corrected as follows:

We use SR measurements at the Eskdalemuir ==Geomagnetic== Observatory (ESK) in the UK operated by the British Geological Survey (Beggan and Musur, 2018; Musur and Beggan, 2019) between 2013 and 2021.

**[Technical Comment 3]**

**Both Line 192 and Line 257 use similar phrasing that is difficult to follow: "occurs in a decrease" and "occurs in a huge decline". Simplifying the wording here would make these sentences flow better, e.g. "…global lightning activity decreased by ~10% from 2019 to 2020 (Fig. 2a)." and "…global lightning declined by ~10% from 2019 to 2020."**

Reply:

Thank you for pointing this out. Following the suggestion, we have revised the corresponding expressions:

Lines 197–198: Noteworthily, SR and ISS LIS observations conformably show that global lightning activity ==decreased by ~10% from 2019 to 2020== (Fig. 2a).

Line 266: As mentioned in Section 3.1, global lightning ==declined by ~10% from 2019 to 2020==.

**[Technical Comment 4]**

**Supplement: There is a typo in first sentence of Text S1: "mothed"- maybe should be method?**

Reply:

Thanks for your kind comment. It has been corrected as follows:

We evaluate the performance of SR observation ==method== with the space-based tropospheric monitoring instrument (TROPOMI) $NO_2$ observations by cloud-slicing

technique (Figures S3a-d and S4a-b).